# Computed Tomographic Analysis of Mandibular Tori and Their Relationship to Remaining Teeth

**DOI:** 10.3390/diagnostics15040414

**Published:** 2025-02-08

**Authors:** Kai Shibaguchi, Kenzo Morinaga, Yuki Magori, Toyohiro Kagawa, Takashi Matsuura

**Affiliations:** 1Section of Fixed Prosthodontics, Department of Oral Rehabilitation, Fukuoka Dental College, 2-15-1 Tamura, Sawara, Fukuoka City 814-0175, Japan; 2Department of Oral Implantology, Osaka Dental University, 1-5-17 Otemae, Chuo, Osaka City 540-0008, Japan; 3Section of Imaging Diagnosis, Department of Diagnosis/Systemic Management, Fukuoka Dental College, 2-15-1 Tamura, Sawara, Fukuoka City 814-0175, Japan

**Keywords:** mandibular tori, computed tomography, morphological analysis, CT value, autogenous bone graft material, remaining teeth

## Abstract

**Objectives:** Mandibular tori (bilateral, asymptomatic, lingual mandibular protuberances) often remain untreated. When considering surgical resection, understanding the shape, size, and position of the tori at the bone level is crucial. However, collecting accurate information regarding these characteristics is challenging in cases where the oral mucosa is prominent on the floor of the mouth. **Methods:** We conducted retrospective surveys at Fukuoka Dental College Medical and Dental General Hospital using computed tomographic (CT) image analysis software (Simplant Pro 18.0). The specific aims of this study were to evaluate the appearance rate of mandibular tori by morphological type, size, location, and CT values and their relationship with the remaining teeth in 1176 patients. These patients underwent simple mandibular CT tomography. We used *t*-tests to analyze the data. **Results:** Approximately 10% of the identified tori were pedunculated and difficult to diagnose through visual inspection alone. In all the age groups > 30 years, patients with mandibular tori had a lower rate of tooth loss and a higher rate of remaining occlusal support than healthy subjects. The mean CT value of the mandibular tori was >1350 Hounsfield units (HU). **Conclusions:** These findings provide insights into future classification and treatment planning for mandibular tori, including that in regard to mandibular ridge resection and factors that may contribute to mandibular torus development or progression, and validate the use of excised bone tissue as a bone graft material.

## 1. Introduction

Mandibular tori are asymptomatic bulges of the jawbone that commonly occur between the canines and premolars on the lingual side of the mandible [1,2,3,4]. The prevalence of mandibular tori varies by region and age, with incidences of 5.2–18.8% in Europeans [5,6], 12.1% in Africans [7], and 9.2–29.9% in Asians [5,8], with most cases affecting people aged 30–50 years. In Japan, 26.5% of the studies on mandibular tori targeted people over 60 years of age, whereas 58.3% of the studies targeted young dentate adults [9,10]. Mandibular tori usually present with few symptoms; therefore, treatment is usually limited to follow-up and observation. However, depending on their size, shape, and area of occurrence, they may interfere with dental treatment and daily living [11,12]. Specifically, mandibular tori can interfere with the design or removal of mandibular dentures [13,14] and plaque control. Mandibular tori have also been reported to cause sleep apnea [15,16]. Surgical resection was performed in these patients. Mandibular torus is usually diagnosed via inspection and palpation of the oral cavity [17,18]. X-ray examinations are also commonly used but are not very effective. Indeed, studies and classifications of the morphology of the mandibular tori by Jainkittivong et al., Eggen et al., and Monica et al. made diagnoses based on inspection and palpation during intraoral examinations [19,20,21].

However, the mandibular tori are typically covered by oral mucosa. Therefore, when a mandibular torus is located beneath the mucosa lining the floor of the mouth or develops into a pedunculated shape, accurately characterizing the protuberance becomes difficult. Incorrect determination of the torus shape may lead to inaccurate information being obtained during preoperative examination prior to surgical resection [22,23] and treatment planning or improper setting of the incision line. In addition, if the mandibular torus is polycystic and pedunculated, the presence of oral mucosa may lead to the misidentification of the type of tooth at the site of development. This may result in the acquisition of inaccurate information when investigating the relationship between mechanical factors with and without occlusal support. Past research has not yet yielded a comprehensive classification of the exact location, shape, and size of a mandibular torus through an analysis of data obtained by constructing 3D images; measurements of the bone density of a mandibular torus and a comparison of these measurements with those for healthy cortical bone; and the correlation between the appearance of the mandibular torus and remaining teeth, in addition to the occlusal relationship according to tooth type in a large, population-based study. In this study, we utilized the computed tomographic (CT) image analysis software SIMPLANT^®^ (Simplant Pro 18.0) developed by Dentsply Sirona, which offers accurate three-dimensional image construction and detailed image resolution. This software, frequently used in treatment planning for dental implants, can recognize the three-dimensional morphology of hard tissue and calculate bone volume and density [24,25]. This study is expected to fill in the gaps regarding information such as the location, shape, and size of mandibular tori, especially for those located deep in the floor of the mouth or with complex shapes, because the authors of previous studies have only used visual inspection, palpation, and CT scans to diagnose the condition. This insight will enable the development of clearer and safer treatment protocols, such as evaluations of the difficulty of treating a mandibular torus, the locations and sizes of gingival incisions, and the location of bone cutting. In addition, while previous studies have only examined the relationship between mandibular tori and the number of missing teeth, this study will clarify the relationship between the number of teeth and the type of tooth and the relationship with teeth in the areas where a mandibular torus is likely to occur, providing highly accurate information about the possibility of mechanical factors such as occlusal contact being involved in the development of mandibular tori in relation to the remaining teeth.

The specific objectives of this study were to investigate the morphological type, exact size, vertical position, and occurrence rate of mandibular tori based on CT values—factors that are difficult to accurately diagnose through visual inspection of the oral cavity alone—and compare the survival rates of teeth and the occlusal support relationship by tooth type.

## 2. Materials and Methods

### 2.1. Participants

Data were obtained from patients who underwent plain mandibular computed tomography (CT) at the Department of Radiology of Fukuoka Dental College Hospital between 1 April 2015, and 31 March 2020. A total of 2158 people were eligible, and 147 of these patients were randomly selected from the teenage group, which had the lowest number of patients. Another 147 individuals were selected from each of the other age groups to ensure that there was a consistent sample size across all age groups. Those 80 years old and older were placed in a separate group, and a total of 1176 people were surveyed. Medical records and CT images from our hospital were used for this survey. To identify and evaluate the mandibular tori, clinical evaluations were made based on intraoral findings acquired by physicians, and radiographic interpretation reports were made based on findings acquired by radiology technicians. The survey contents included age at the time CT imaging was performed, sex, the presence or absence of a mandibular torus, and detailed information on the mandibular torus (the site of occurrence, thickness, shape, volume, and bone density). Detailed information on the mandibular ridge was obtained by converting the files into three-dimensional CT images using SIMPLANT^®^ (Simplant Prp 18.0, Dentsply Sirona) (Figure 1). As per the inclusion criteria, we included patients who were judged to have a mandibular morphology and bone density within the normal range and who underwent CT imaging to diagnose pathologies that did not involve significant anatomical changes and formulate treatment plans to achieve the following: (1) obtain accurate diagnoses before implant placement surgery, (2) understand the positional relationship between the impacted mandibular wisdom teeth and the mandibular canal, (3) diagnose residual apical tooth lesions and cysts and/or (4) identify the presence and extent of fracture lines at the time of trauma, and (5) for the differentiation of other dental diseases. We excluded the following patients: (1) those whose continuity of normal cortical bone was lost due to trauma, cysts, tumors, or tumorous bone swelling in the mandible; (2) those with a history of surgical jawbone reshaping and tumor resection in the mandible; (3) those with a history of bone disease, such as osteogenesis imperfecta or osteosclerosis; (4) those taking or who had once taken osteoporosis medication; (5) those who refused to participate in the study; and (6) those judged to be unsuitable research subjects.

### 2.2. Classification of Morphology and Size

The morphologies of the mandibular tori were classified as types I to IV, according to the classification method used in previous studies [1,21,22] (Figure 2). The thickness of each torus was measured from the outline of the normal cortical bone to the crest of the ridge (Figure 3) and was classified into the following categories in accordance with previous studies [17,22,26]: <3 mm, mild; 3–6 mm, moderate; and 6 mm or more, severe. When a mandibular torus was polycystic, it was classified based on its thickness.

### 2.3. Bone Density Assessment

The CT value for each mandibular torus was the mean of the CT values measured at the base, center, and top of the bone (Figure 4). Three points on the cortical bone, without any obvious bulging, were arbitrarily selected and measured, and the mean of these values was used as the control value. The apical positions of the teeth were demarcated on the cross-sectional plane to provide a reference for the vertical position of the tori. The bone on the coronal side of this line was considered the alveolar bone, and the bone on the apical side was considered the mandibular bone.

### 2.4. Definition of Remaining Teeth and Occlusal Support

Regarding the presence or absence of remaining teeth, the patients whose roots remained, regardless of the condition of the crown, were considered to have remaining teeth. Implants and fixed bridge pontics were classified as “no remaining teeth.” Molars that underwent root resection, hemisection, trisection, or root separation were also classified as “remaining teeth.” As for the definition of occlusal relationships, cases in which the same type of tooth was found to remain in the upper and lower jaws on the same side on the left and right were defined as an “occlusal support relationship.”

### 2.5. Statistical Methods

The following statistical methods were used to analyze the collected data: The independent samples *t*-test was used to handle and compare continuous data such as the number of remaining teeth and occlusal support values in different populations of patients with and without mandibular torus. Test assumptions were checked using the Shapiro–Wilk test. SPSS was used to perform the analysis. The results are reported as means and standard deviations, and the significance level was set at *p* < 0.05.

## 3. Results

### 3.1. Prevalence of Mandibular Tori

Of the 1176 patients, 334 (28.4%) had mandibular tori and 79.5% had bilateral tori. The prevalence of mandibular tori increased and then decreased with age, peaking in patients in their 40s (Table 1). Specifically, tori were observed in 49.7% of people in their 40s, constituting the highest incidence rate, followed by 40.1% of people in their 50s, and 39.5% of people in their 30s. These incidence rates of mandibular tori are close to those reported in previous studies [9,10]. In previous studies, the most common age of onset was from 30 to 50 years of age [5,7], a range that is also close to the results of our study.

### 3.2. Frequently Occurring Site of Mandibular Tori

Of all of the tori, 84.9% appeared in the canine–premolar region (Figure 5). The incidence was the highest in the first premolar region, accounting for 32.3% of all cases. Regarding the vertical position, more than 80% of the lesions were localized in the alveolar bone across all age groups. Previous studies reported that the incidence rate was high for the region from the canines to the premolars [17,19,27], a finding consistent with the results of our study.

### 3.3. Appearance Rate of Mandibular Tori by Morphological Type

Regarding torus morphology, the monocystic and stemless types were the most common (57.9%), 5.47% of mandibular tori were of the monocystic and pedunculated types, and 5.17% were multivesticular and pedunculated (Figure 6). In total, 10.5% of the mandibular tori were of the pedunculated type, which is often difficult to accurately diagnose through a visual inspection of the oral cavity alone. Regarding size classification, >50% of cases across all age groups were classified as mild. Among the patients in their 30s, 50s, and 60s, >40% of cases were classified as moderate. The most prominent mandibular torus measured 10.5 mm from the base to the point of maximum protrusion. Investigations of torus volume revealed a maximum volume of 1826 mm^3^. Although previous studies have mentioned the types of morphology [1,19,21], they have not investigated detailed occurrence rates classified by each morphology, constituting a highly novel aspect of this study.

### 3.4. Bone Density of Mandibular Tori and Healthy Cortical Bone

When compared with the control group, the mean CT value of the mandibular tori was approximately 200 HU (Hounsfield units, used in CT imaging to measure the X-ray attenuation coefficient and, subsequently, bone density) lower in individuals up to 80 years of age (i.e., those in their 10s to 70s). This difference was significant (Figure 7). Across all age groups, the mandibular tori had an average bone density greater than 1350 HU. In a previous study [24], the CT values of the mandibular tori themselves were reported to be in the range of approximately 1200–1900 HU, similar to the values in our study. However, the true purpose of this study was to measure the thickness of the mandibular tori, not to calculate the CT values, and previous studies did not compare these values with the CT values of healthy cortical bone. Therefore, this comparison is a novel aspect of our study.

### 3.5. Comparison of Mandibular Tori and Tooth Loss Rate

In all age groups (>30 years), patients with mandibular tori had a lower rate of tooth loss than the healthy subjects. A statistically significant difference was observed, especially in the >50s, 70s, and 80s and older groups (Figure 8). Comparing the percentage of missing teeth by tooth type reveals that the percentage of missing teeth for all tooth types was lower for the patients with a mandibular torus compared to healthy subjects, especially those in their 40s and older. Similarly, when comparing the area from the canine to the premolar region, which is the most common site of a mandibular torus, the proportion of missing teeth was lower in patients with a mandibular torus than that observed in healthy subjects in all age groups. Statistically significant differences were observed in the 50s, 70s, and 80s and older groups (Figure 9). These relationships did not differ significantly between the sexes. Previous research found that people with a mandibular torus had a significantly higher rate of remaining teeth [6]. Our study shows a trend similar to the previous study, and we also obtained similar results when comparing only the canine to the premolar area, the latter being the most common area for a mandibular torus, indicating a more substantial relationship between mandibular tori and remaining teeth.

### 3.6. Comparison of Mandibular Tori and Occlusal Support

Among patients with mandibular tori, occlusal support relationships were maintained for all tooth types in all groups of individuals aged 30 years or older. In all those >30 years, the rate of remaining occlusal support was higher for patients with a mandibular torus than observed in healthy subjects, and statistically significant differences were observed in the 30s, 60s, and older age groups (Figure 10). Similarly, in the canine–premolar region, a comparison of the proportion of patients who maintained occlusal support showed that patients in their 30s and older had better occlusal support than healthy subjects, with a statistically significant difference observed for those in their 60s and older (Figure 11). A previous study reported that there was a correlation between the wear of premolars due to occlusal contact and the frequency of mandibular torus appearance; however [28], in cases where there was no occlusal contact on the premolars, a mandibular torus did not appear. This may be one of the studies supporting our finding that the occurrence rate of mandibular tori was significantly higher when there was occlusal support. Among the patients with a mandibular torus, only one patient in the 60s group had an occlusal support number below five for both their upper and lower jaws, and no other cases were observed. Edentulism in both the upper and lower jaws was found in 6.1% of healthy individuals in their 60s, 11.1% of the patients were in their 70s, and in 17.2% of those over 80 years of age, but no mandibular torus cases were observed in any of these age groups. These results differ from those of a previous study [14] that investigated the incidence of mandibular tori in edentulous individuals.

## 4. Discussion

This study had two main objectives. The first objective was to obtain accurate information on the location, shape, size, and bone density of mandibular tori using CT analysis software, allowing us to discuss the importance of diagnosis at the bone level in mandibular torus resection and its impact on treatment protocols and to obtain knowledge on the characteristics of mandibular tori. The main motivation for this objective is the difficulty of accurately characterizing the mandibular ridge when it is beneath the mucous membrane of the floor of the mouth; therefore, detailed characterization via three-dimensional analyses is considered necessary. Pedunculated mandibular tori can be excised relatively easily by resecting them at the base. However, if the torus is not pedunculated or it is between the alveolar bone area and the body of the mandible, a larger incision surface area is required to fully resect the torus, increasing the technical difficulty of the procedure and the burden on the patient. The extent of gingival mucosa required for resection varies from case to case. Therefore, the results of our study emphasize the importance of using CT analysis software for preoperative examinations when performing mandibular torus resection, and it is believed that, at the preoperative diagnosis stage, treatment protocols can be made more detailed and clearer, including with respect to evaluating the technical difficulty of resecting the mandibular ridge located in the deep area of the floor of the mouth, identifying the areas of bone to be resected or preserved, determining the extent of the mucosal incision, and providing detailed explanations to patients according to the difficulty of treatment. In this study, the most protruding mandibular torus was 10.5 mm, and the maximum volume was 1826 mm^3^. Diagnosis with CT image analysis gives one the ability to extract only the mandibular torus in an image and measure its exact size, and the ease of imaging in 3D makes it easier to create a treatment plan for resection. A limitation of this study is that the results may not accurately reflect the condition wherein the mucosa is incised and the bone tissue is exposed due to factors such as the resolution of CT image analysis, artifacts caused by the presence of dental metals, and the impossibility of analyzing soft tissue.

A notable component of this study was that the CT analysis software was applied to over 1000 participants. This allowed us to accurately classify the size and morphologies of the tori and compare the bone density between the torus and healthy bone in the same patient. Jean-Daniel et al. performed a morphological analysis of mandibular tori using micro-CT imaging [29]; however, the sample size was very small (*n* = 16). Conversely, in this study, morphological analysis via CT imaging was performed on 334 patients with mandibular tori. Our larger sample size increased the statistical validity of our findings. In a systematic review of the incidence of mandibular tori conducted by Garcia-Garcia et al. [1], all 13 cited studies’ authors used visual inspection and palpation to diagnose mandibular tori [6,7,18,19,26,27]. Thus, no previous studies have investigated the incidence and common sites of mandibular tori in a large patient pool using CT analysis software. Furthermore, in our image analysis pipeline, we extracted and calculated only the volumes of the mandibular tori. In previous studies, most methods for estimating torus size involved measuring the distance to the maximum bulge point of the torus [10,13,17,21,29]. In this study, in addition to the measurement method, mandibular tori were digitally extracted, and the volume of this portion was calculated. Therefore, this method proved valid for determining the scale of the entire mandibular torus. CT image analysis can also be used to accurately determine the bone density using CT values [25,30]. Youna et al. investigated the thickness and CT values of mandibular tori using CT image analysis [24]. However, in our study, we compared the densities of mandibular tori and healthy bones. Therefore, from a radiological perspective, we could discern whether each mandibular torus was merely a swelling of the cortical bone or a tumor with a distinct tissue structure.

The second objective of this study was to characterize the oral environment in which a mandibular torus is likely to appear, as well as to conduct a detailed evaluation of whether there is a relationship between the presence or absence of a mandibular torus and the presence of remaining teeth and occlusal support, in order to seek evidence showing whether or not mechanical factors such as biting force affect the occurrence and progression of mandibular tori. In this study, mandibular tori occurred within the canine–premolar range, and over 80% of cases occurred within the range of the alveolar bone. In addition, the rate of tooth loss was significantly lower in mandibular torus patients in their 40s and older, suggesting that the occurrence of mandibular tori may be closely related to the presence of the remaining teeth and alveolar bone. Consequently, the conclusions of previous studies [6,29] were confirmed more concretely through this survey. The older the age group, the greater the difference in the proportion of missing teeth between patients with a torus and healthy subjects. This may be because the proportion of people with missing teeth was low in the younger age groups; therefore, the difference in the proportion of missing teeth between torus patients and healthy subjects was small, and the proportion of people with missing teeth generally increased with age. The rate at which the occlusal support relationship was maintained was significantly higher in patients with a mandibular torus than in healthy subjects, suggesting a relationship between the development of mandibular tori and occlusal support. Previous research has shown that the occurrence of a mandibular torus is significantly related to the wear of the remaining teeth and the position of the occlusal contacts [9], evidence that may support the findings of this study.

The novelty of this study is that it examined the relationship between the appearance of mandibular tori and the remaining teeth when examining all tooth types as the target area and when examining only the canine–premolar area, where a mandibular torus is most likely to appear. As a result, similar trends were observed when examining the entire oral cavity; moreover, when examining only the canine–premolar area, a statistically significant difference was observed. This tendency was observed both when comparing the mandibular torus appearance rate with the rate of tooth loss and when comparing the mandibular torus appearance rate with the presence of occlusal support. In other words, it seems that similar results were obtained when examining the presence or absence of remaining teeth in the area where a mandibular torus is most likely to appear and when examining the presence or absence of remaining teeth in the entire oral cavity. This finding can also be explained by the importance of examining the number of remaining teeth and the presence or absence of occlusal relationships in the entire oral cavity. Therefore, investigating the rate at which mandibular tori appear and the number of remaining teeth in the entire oral cavity will support the findings of previous studies that have pointed out the possibility of a relationship between mandibular tori and remaining teeth, the occlusal relationship, and biting force [6,31]. Recent studies have explored the relationship between parafunctional activities, such as bruxism, and the incidence of mandibular tori [8,32]. Previous reports have shown that tooth loss prevents the stimulation of the alveolar bone, causing a loss of bone width and height [33,34]. A follow-up study on edentulous patients showed that the degree and angle of vertical alveolar bone resorption are greater in the mandible than in the maxilla [35].

Excised mandibular tori are usually discarded. However, in recent years, they have also been used for the pretreatment of prosthetics at sites scheduled for implantation surgery [36,37,38,39,40]. In addition, some case reports have described the use of excised tori as autogenous bone grafts to address vertical alveolar bone density, which is important for determining the extent to which bones recover from autogenous bone grafting. Histological evaluation of mandibular tori has been reported to help indicate potential autologous bone graft material [41]. However, a detailed radiological investigation of mandibular tori has not yet been conducted. The Hounsfield unit (HU) is a radiographic measure of bone density [30]. In this study, the CT values of the mandibular tori were significantly lower than those of the healthy cortical bone in each age group; however, the average value was over 1350 HU. This value corresponds to “Grade: D1” (1250 HU or more), considered the most predictable according to Misch’s bone quality classification [30]. Therefore, the findings suggested that the mandibular tori had densities that would be considered normal when assessed via radiography. This study may provide some scientific evidence, at least from a radiological standpoint, in favor of the application of mandibular tori as a bone graft material. A limitation of this study is that it is based only on radiological bone density, so it is not possible to make a comprehensive statement about the quality of the mandibular torus as a bone graft material. Based on the bone density values obtained using X-rays, the mandibular tori had bone quality at least comparable to that of the healthy cortical bone, suggesting that they could be used as an alternative to areas that have been used as donor material for bone grafts up to now. In addition to bone density determined via X-rays, it is expected that further comprehensive investigations will be required in the future to investigate the types and composition ratios of cells that make up mandibular tori, as well as their bone differentiation ability, in order to determine the bone quality that is required.

This study’s retrospective design, in which we used data collected from past patient records and CT images, may have introduced patient selection bias and limited the reliability of the data collection methods and patient characteristics. In addition, this study only included patients who underwent plain mandibular CT scans, limiting its generalizability and excluding cases of mandibular tori diagnosed using methods other than CT imaging. Thus, the results do not reflect the true prevalence of mandibular tori. Furthermore, a monomodal diagnostic approach does not consider other diagnostic methods or clinical evaluations that may provide complementary information. In addition, in cases in which CT scans were performed within a short period after the loss of the remaining teeth located at the site where a mandibular torus developed, the mandibular torus was considered to have developed at the site where the teeth were missing. One of the uncertainties in this study design may be that occlusal support was counted if the same type of tooth was found to be present in the upper and lower jaws on the same side, regardless of the crown morphology of the remaining teeth or whether there was actual occlusal contact. Therefore, the validity of analyzing the relationship between the presence of a mandibular torus and of remaining teeth or the presence or absence of occlusal support may have been reduced. However, due to the limited number of studies employing this research design, this information is important with respect to considering the possibility that mechanical factors, such as the presence of remaining teeth and occlusal support, are involved in the development and maintenance of a mandibular torus. Furthermore, the number of remaining teeth and the presence or absence of occlusal support in the canine–premolar area, which is the most common site for a mandibular torus, represents a highly novel aspect of this study.

## 5. Conclusions

This study had two main objectives. A large-scale survey of mandibular tori conducted using CT analysis software has provided detailed morphological classifications, sizes, and bone density evaluations via CT image analysis that have not been available until now. This has revealed the possibility of further clarifying the treatment protocols for mandibular torus resection. In addition, one aspect of the bone quality of the mandibular tori was obtained from a radiological perspective, thus narrowing down the content to be further examined in order to evaluate the suitability of tori as bone graft materials. Furthermore, it was found that the rate at which mandibular tori appear is related to the presence or absence of remaining teeth and occlusal support in the area where a mandibular torus is likely to occur, elucidating content that has not been clearly explained in past studies, such as the involvement of mechanical factors in the occurrence of a mandibular torus and the possibility of the weakening or disappearance of a mandibular torus due to tooth loss.

## Figures and Tables

**Figure 1 diagnostics-15-00414-f001:**
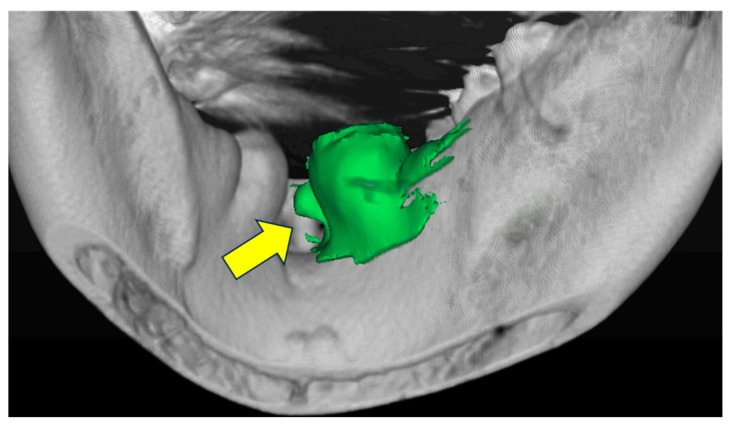
Identification and analysis of mandibular tori using computed tomography analysis software. The arrow points to the mandibular torus identified. This information allowed for the evaluation of morphology, size, bone density, etc., at specific locations.

**Figure 2 diagnostics-15-00414-f002:**
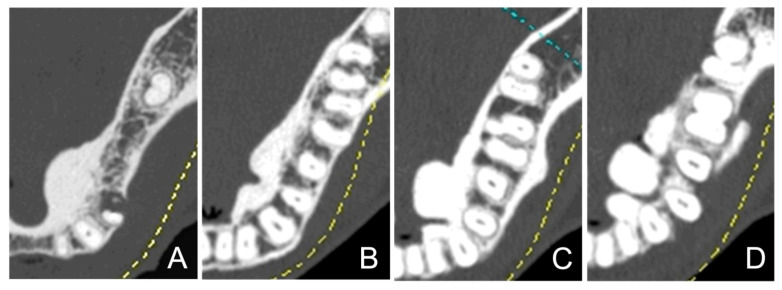
Morphological classification of mandibular tori according to computed tomography images. Mandibular tori were categorized into four types based on their morphologies. The examples are displayed in sequence from Type I to Type IV, starting from the left. Type I: monocystic and stemless (**A**). Type II: multivesicular and stemless (**B**). Type III: monocystic and pedunculated (**C**). Type IV: multivesicular and pedunculated (**D**).

**Figure 3 diagnostics-15-00414-f003:**
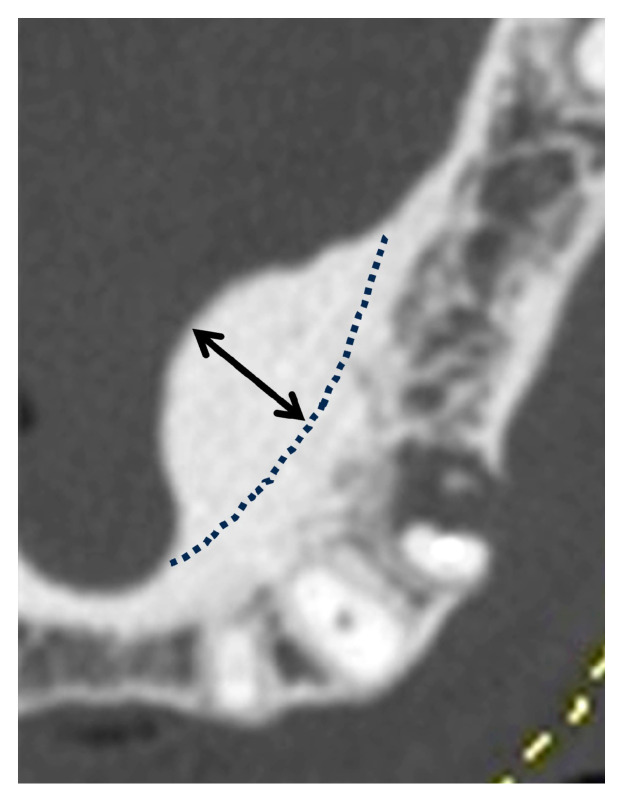
Measurement points used to determine the sizes of the mandibular tori. The distance from the outline of the normal cortical bone (represented by the dotted line in the figure) to the point of maximum protrusion (indicated by the double-headed arrow) was measured. This distance indicates the size of a mandibular torus.

**Figure 4 diagnostics-15-00414-f004:**
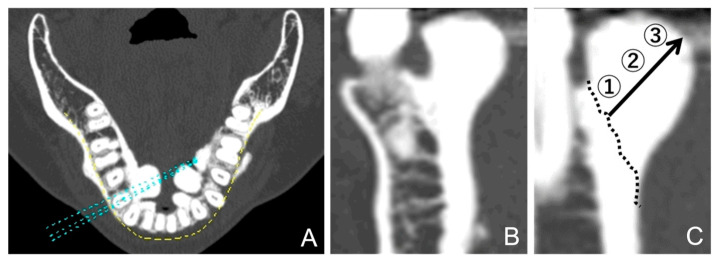
Results regarding the computed tomography (CT) values calculated for the mandibular tori. (**A**) A panoramic curve set in the horizontal section of the CT image. (**B**) A cross-sectional surface in the vertical direction. (**C**) After setting the boundaries of the expected anatomical cortical bone (dotted line), three areas were defined: 1. the base area, 2. the central area, and 3. the maximum bulge area.

**Figure 5 diagnostics-15-00414-f005:**
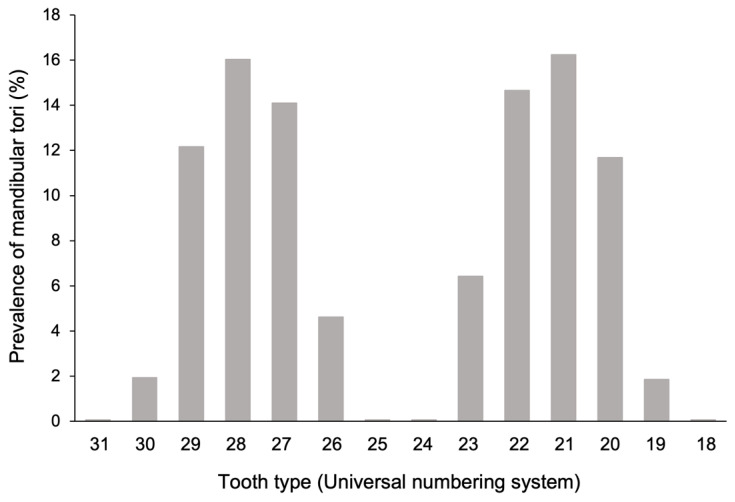
The prevalence of mandibular tori compared by tooth type area. The values on the horizontal axis indicate the types of teeth, presented using the universal numbering system. The vertical axis shows the prevalence (%) of mandibular tori.

**Figure 6 diagnostics-15-00414-f006:**
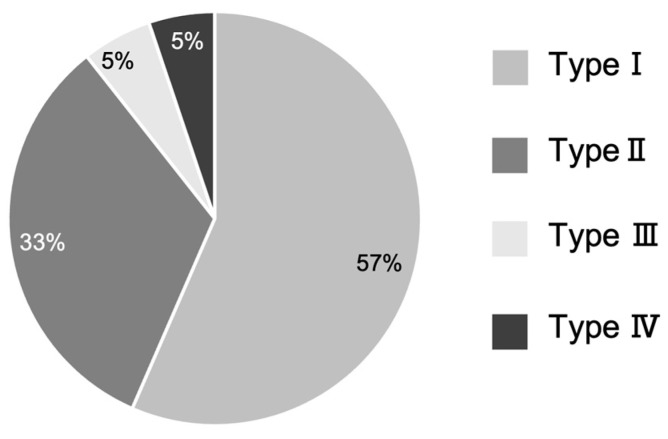
Breakdown of mandibular tori by morphological type. Type I indicates the monocystic and stemless type. Type II indicates the multivesicular and stemless type. Type III indicates the monocystic and pedunculated type. Type IV indicates the multivesicular and pedunculated type.

**Figure 7 diagnostics-15-00414-f007:**
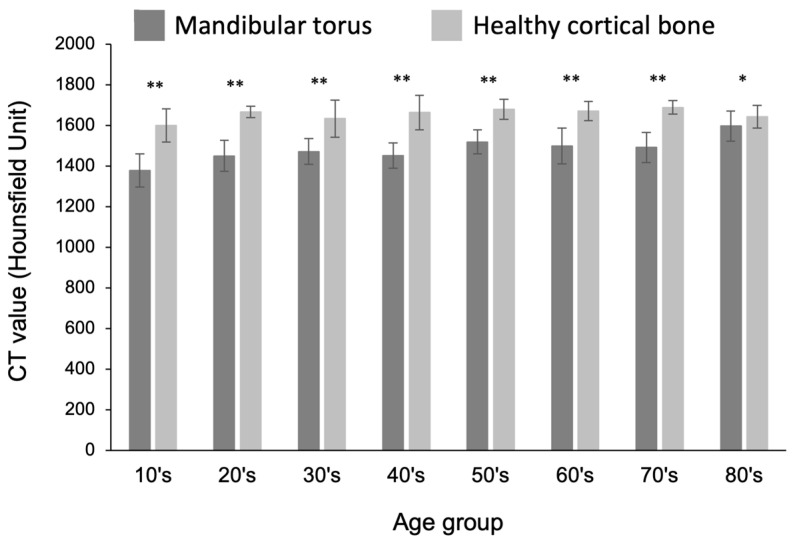
Comparison of bone density between mandibular tori and healthy cortical bone in each age group. The horizontal axis represents the different age groups, while the vertical axis displays the CT values (in Hounsfield units). A *t*-test was used for statistical analysis. An asterisk designates network metrics with a significant difference (“*”: *p* < 0.05, “**”: *p* < 0.01).

**Figure 8 diagnostics-15-00414-f008:**
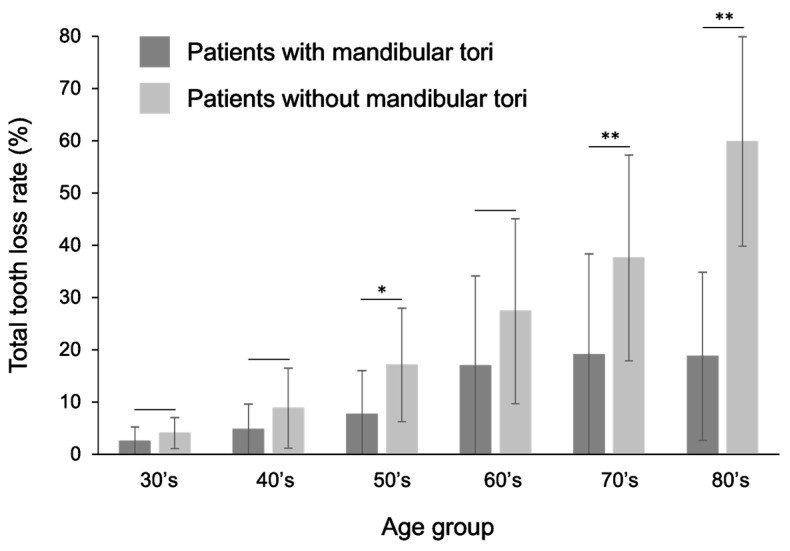
Comparison of tooth loss rates between patients with mandibular tori and healthy subjects. The rate of tooth loss was lower in patients with a mandibular torus than that observed in healthy individuals in all the over-30 age groups. Statistically significant differences were observed, especially for those in their 50s, 70s, and 80s. A *t*-test was used for statistical analysis. An asterisk designates network metrics with a significant difference (“*”: *p* < 0.05, “**”: *p* < 0.01).

**Figure 9 diagnostics-15-00414-f009:**
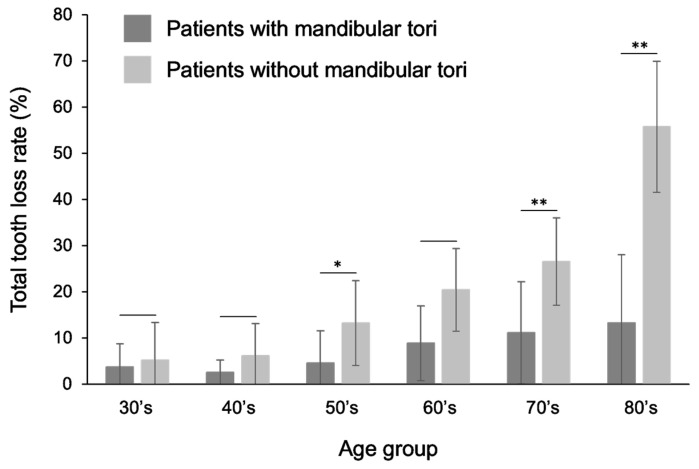
Comparison of tooth loss rates between patients with a mandibular torus and healthy subjects, focusing only on the canine to premolar area. The rate of tooth loss was lower in patients with a mandibular torus than that observed in healthy individuals in all those over 30 years old. Statistically significant differences were observed, especially for those in their 50s, 70s, and 80s. A *t*-test was used for statistical analysis. Asterisks denote network metrics with a significant difference (“*”: *p* < 0.05, “**”: *p* < 0.01).

**Figure 10 diagnostics-15-00414-f010:**
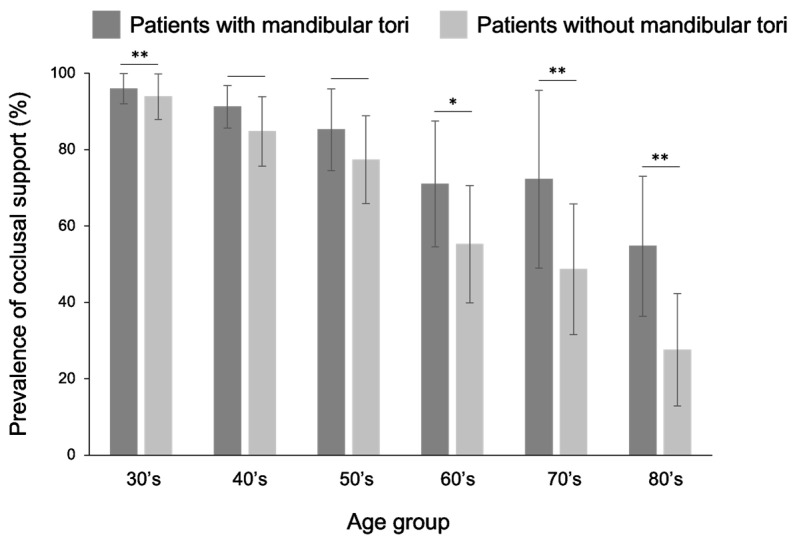
Comparison of the rate of occlusal support between patients with a mandibular torus and healthy subjects. The rate of tooth loss was lower in patients with a mandibular torus than that observed in healthy individuals in all those over 30 years old. Statistically significant differences were observed, especially for those in their 50s, 70s, and 80s. A *t*-test was used for statistical analysis. An asterisk denotes network metrics with a significant difference (“*”: *p* < 0.05, “**”: *p* < 0.01).

**Figure 11 diagnostics-15-00414-f011:**
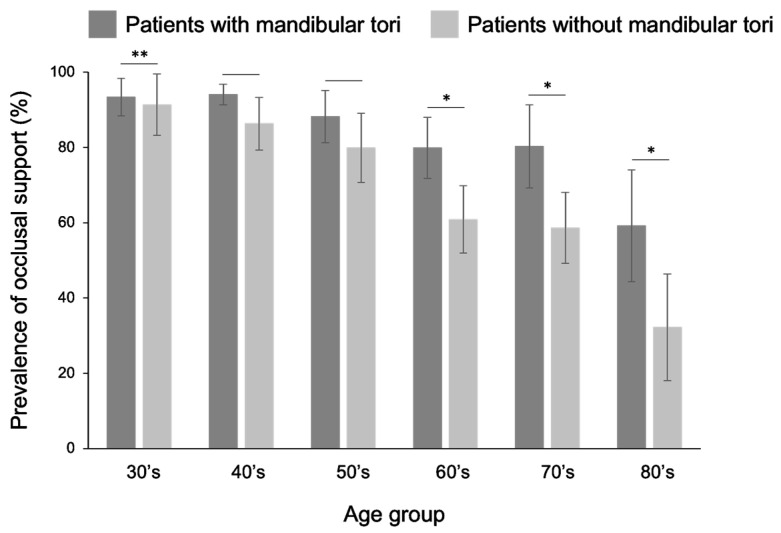
Comparison of the rate of occlusal support between patients with a mandibular torus and healthy subjects, focusing only on the canine–premolar area. The rate of tooth loss was lower in patients with mandibular torus than that observed in healthy individuals in all those over 30 years old. Statistically significant differences were observed, especially for those in their 50s, 70s, and 80s. A *t*-test was used for statistical analysis. An asterisk denotes network metrics with a significant difference (“*”: *p* < 0.05, “**”: *p* < 0.01).

**Table 1 diagnostics-15-00414-t001:** Prevalence of mandibular tori in each age group.

Age Group	Prevalence (%)	Number of Subjects
10s	8.2	147
20s	18.4	147
30s	39.5	147
40s	49.7	147
50s	40.1	147
60s	33.3	147
70s	18.4	147
80s and above	19.7	147

The table presents the prevalence (%) of mandibular tori in the 147 individuals in each age group.

## Data Availability

The data presented in this study are available upon request.

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
