# Peer review of "Computed Tomographic Analysis of Mandibular Tori and Their Relationship to Remaining Teeth"

_diagnostics, 2025, doi:10.3390/diagnostics15040414_

Round 1

Reviewer 1 Report

Comments and Suggestions for Authors

Dear Authors,

Thank you for the opportunity to review your manuscript on the computed tomographic analysis of mandibular tori and its relationship to remaining teeth. Your research addresses an important and clinically relevant topic with the potential to contribute valuable insights into oral health diagnostics and treatment planning. While the study presents interesting findings, several areas require further clarification and improvement to enhance the manuscript's scientific rigor and clinical applicability.

Introduction:

·       The introduction lacks a clear articulation of the research gap and objective.

·       Recommendation: Provide a stronger statement on how this study fills an existing gap in the literature and the specific hypotheses  tested.

Materials and Methods:

·       The inclusion and exclusion criteria require further clarification, especially regarding potential confounders.

·       Recommendation: Clearly state how patients with varying degrees of dental health were controlled to avoid bias.

·       Statistical methods are mentioned but lack sufficient detail.

·       Recommendation: -If t-tests were used, specify whether they were independent samples t-tests or paired samples t-tests.

               -Explain why a particular statistical test was chosen based on the nature of the data (e.g., continuous vs. categorical data, normal distribution, homogeneity of variance).

              -Mention whether assumptions for the tests were checked (e.g., normality tested using the Shapiro-Wilk test, variance homogeneity tested with Levene's test).

            - Indicate which statistical software was employed to perform the analysis (e.g., SPSS, R, Python, SAS), as this helps with the reproducibility of the study.

            - Explain how the results were reported (e.g., means and standard deviations, confidence intervals, effect sizes for group comparisons).

Results:

•The results section is descriptive but lacks comprehensive interpretation.

Recommendation: Add more context to the findings by comparing them to previous studies and discussing their potential clinical implications.

Discussion:

·       The discussion is repetitive in some areas and lacks critical evaluation of limitations.

Recommendation: Condense redundant information and provide a more balanced discussion of the study's strengths and weaknesses.

·        The potential clinical applications of using mandibular tori as autogenous bone graft material require further exploration.

Recommendation: Expand on how the findings could impact future treatment protocols.

Conclusion:

·       The conclusion section restates results without highlighting broader implications.

Recommendation: Summarize the main findings concisely and suggest future research directions.

References

·       Some citations are outdated and may not reflect the latest advancements in the field.

Recommendation: Update the references to include more recent studies to support claims, if possible.

Ethical Considerations:

·       The section on informed consent is vague.

·       Recommendation: Clarify how patient confidentiality was ensured and whether written informed consent was obtained.

Overall Recommendation: Major revisions are required to improve methodological clarity and data interpretation. Once these revisions are implemented, the study has the potential to make a significant contribution to the field of dental radiology and prosthodontics.

Author Response

Comments 1: The introduction lacks a clear articulation of the research gap and objective.

·Recommendation: Provide a stronger statement on how this study fills an existing gap in the literature and the specific hypotheses tested

Response 1: Thank you for pointing this out. Therefore, I have provided a stronger and clearer explanation of how this study fills existing gaps in the literature and the specific hypotheses tested. This change was made to page 2, lines 56-66 and 70-87 of the revised manuscript.

Comments 2: The inclusion and exclusion criteria require further clarification, especially regarding potential confounders.

· Recommendation: Clearly state how patients with varying degrees of dental health were controlled to avoid bias.

Response 2: Thank you for pointing this out. Therefore, we provided further clarification of the inclusion and exclusion criteria, especially regarding potential confounders. This change was made to page 3, lines 148-152 and 156-161 of the revised manuscript.

Comments 3: Statistical methods are mentioned but lack sufficient detail.

· Recommendation: -If t-tests were used, specify whether they were independent samples t-tests or paired samples t-tests.

               -Explain why a particular statistical test was chosen based on the nature of the data (e.g., continuous vs. categorical data, normal distribution, homogeneity of variance).

              -Mention whether assumptions for the tests were checked (e.g., normality tested using the Shapiro-Wilk test, variance homogeneity tested with Levene's test).

            - Indicate which statistical software was employed to perform the analysis (e.g., SPSS, R, Python, SAS), as this helps with the reproducibility of the study.

            - Explain how the results were reported (e.g., means and standard deviations, confidence intervals, effect sizes for group comparisons).

Response 3: Thank you for pointing this out. Therefore, we have described our statistical methods in detail. This change was made to page 5, lines 219-224 of the revised manuscript.

Comments 4: The results section is descriptive but lacks comprehensive interpretation.

Recommendation: Add more context to the findings by comparing them to previous studies and discussing their potential clinical implications.

 Response 4: Thank you for pointing this out. Therefore, we compared our results with previous studies and included a comprehensive interpretation and discussion of potential clinical implications. This change was made to page 5-8, lines 231-233, 240-251, 265-267, 277-282, 299-303 and 325-330 of the revised manuscript.

Comments 5: The discussion is repetitive in some areas and lacks critical evaluation of limitations.

Recommendation: Condense redundant information and provide a more balanced discussion of the study's strengths and weaknesses.

 Response 5: Thank you for pointing this out. We agree with this comment. Therefore, we have condensed redundant information and provided a discussion of the strengths and weaknesses of the studies and a critical evaluation of their limitations. This change was made to page 10-12, lines 352-356, 366-378, 422-427, 462-464 and 713-716 of the revised manuscript.

Comments 6: The potential clinical applications of using mandibular tori as autogenous bone graft material require further exploration.

Recommendation: Expand on how the findings could impact future treatment protocols.

 Response 6: Thank you for pointing this out. We agree with this comment. Therefore, we provide a discussion of how our findings may impact future treatment protocols. This change was made to page 12, lines 691-702 of the revised manuscript.

Comments 7: The conclusion section restates results without highlighting broader implications.

Recommendation: Summarize the main findings concisely and suggest future research directions.

Response 7: Thank you for pointing this out. Therefore, we provide a brief summary of the main findings and suggest future research directions. This change was made to page 12-13, lines 726-775 of the revised manuscript.

Comments 8: Some citations are outdated and may not reflect the latest advancements in the field.

Recommendation: Update the references to include more recent studies to support claims, if possible.

Response 8: Thank you for pointing this out. Therefore, we have updated the references to include more recent studies that support our claims. This change was made to page 13-14, lines 817-818 and 876-882 of the revised manuscript.

Comments 9: 

The section on informed consent is vague.

· Recommendation: Clarify how patient confidentiality was ensured and whether written informed consent was obtained.

Response 9: Thank you for pointing this out. Therefore, we made clear how patient confidentiality was ensured and whether written informed consent was obtained. This change was made to page 13, lines 785-792 of the revised manuscript.

Reviewer 2 Report

Comments and Suggestions for Authors

The introduction is well written and fairly introduces the topic of mandibular tori and its relationship to tooth loss

A suitable question to raised is, why were implants considered as remaining teeth. Indeed tooth were lost and implant was replacing it, so the presence of implant doesnt give a proper correlation of mandibular tori with presence of teeth

the above  then also raises concern about the section 3.5 and 3.6 in results

The above concern is should be discussed in the discussion section, as to why implants were considered as remaining teeth

Author Response

Comment 1: The introduction is well written and fairly introduces the topic of mandibular tori and its relationship to tooth loss

A suitable question to raised is, why were implants considered as remaining teeth. Indeed tooth were lost and implant was replacing it, so the presence of implant doesnt give a proper correlation of mandibular tori with presence of teeth

the above  then also raises concern about the section 3.5 and 3.6 in results.

The above concern is should be discussed in the discussion section, as to why implants were considered as remaining teeth.

Response 1: Thank you for pointing this out. We apologize for any confusion that may have arisen as implants were incorrectly defined as "remaining teeth". Therefore, we have revised implants to "no remaining teeth". This change was made to page 5, lines 213 of the revised manuscript.

Round 2

Reviewer 1 Report

Comments and Suggestions for Authors

I extend my appreciation to the authors for their meticulous revisions and thoughtful contributions. The modifications have significantly enhanced the clarity and comprehensiveness of the content. At this stage, no further changes are required, as the text fully meets the intended academic and professional standards.